# Age-Related Differences in Psychological Distress during the COVID-19 Pandemic

**DOI:** 10.3390/ijerph19095532

**Published:** 2022-05-02

**Authors:** Angelo Rega, Raffaele Nappo, Roberta Simeoli, Mariangela Cerasuolo

**Affiliations:** 1Department of Humanistic Studies, University of Naples Federico II, 80138 Napoli, Italy; roberta.simeoli@unina.it; 2Neapolisanit s.r.l. Rehabilitation Center, 80044 Ottaviano, Italy; raffaele.nappo@neapolisanit.net; 3Associazione Italiana Per L’Assistenza Spastici Onlus Sez Di Cicciano, 80033 Cicciano, Italy; 4Faculty of Medicine, University of Ostrava, 70103 Ostrava, Czech Republic; 5Department of Psychology, University of Campania L. Vanvitelli, 81100 Caserta, Italy

**Keywords:** psychological distress, COVID-19 pandemic, students

## Abstract

While the negative impact of COVID-19 total lockdown on mental health in youth has been extensively studied, findings collected during subsequent waves of the pandemic, in which restrictive rules were more eased, are very sparse. Here, we explore perceived psychological distress during the partial lockdown of the third wave in Southern Italy in a large sample of students, focusing on age and gender differences. Also, we assessed whether attending the type of education could have a protective role on students’ psychological well-being. An online survey was completed by 1064 southern Italian students (age range: 8–19 years; males = 368) from March to July 2021. The survey consists of a set of questions regarding general sociodemographic information as well as several aspects of students’ psychological well-being. Psychological distress was higher in high school students compared to both elementary and middle ones. In addition, we found gender differences, but only in high school students, with females reporting higher psychological distress than males. Finally, our mediation analysis showed a mediated role of face-to-face schooling in the relationship between age and psychological distress. In conclusion, this study highlights age-related differences in psychological distress during the pandemic and the protective role of school in presence for mental health in Italian students.

## 1. Introduction

Psychological distress refers to non-specific symptoms of stress, anxiety and depression [1], which may represent a normal fluctuation in mood or indicate the onset of a major depressive, anxiety, somatization disorder or a variety of other clinical conditions (APA Dictionary of Psychology). This psychological condition refers to the general concept of “maladaptive psychological functioning in the face of stressful life events” [2]. Indeed, psychological distress is more likely to occur in traumatic and stressful situations, especially those that can shatter our sense of security, make us feel helpless and vulnerable and suddenly change our everyday life. One of the traumatic events that we have all been facing for almost two years is the COVID-19 pandemic. In fact, the pandemic emergency and the strategies used to contain its spread are complex chronic psychosocial stressors. In particular, in Italy, from March 2020, very restrictive measures were adopted to prevent the contagion, such as movement restrictions, smart working, closure of non-essential stores and schools of every order and degree, which lasted 2 months before restrictions were relaxed. With new rises of COVID-19 cases during the second (from September 2020) and third wave of the pandemic (from February 2021), the Italian government imposed a partial lockdown regionally graded according to a set of risk parameters including the contagion rates and the pressure undergone by healthcare systems: the higher the risk, the greater the restrictions.

The negative impact of the “first wave” (spring 2020) of the contagion on psychological well-being in the general population is now well documented, including post-traumatic stress symptoms, depression, anxiety and anger (for reviews see: [3,4]). Some recent studies showed persistent effects even during the second wave [5,6,7].

Recently, greater attention has been paid to the effects of the pandemic on children and adolescents. In fact, the changes in their classic daily routine, due to extended school closure, the shutdown of extracurricular activities and social isolation, might represent a risk for both their physical and mental health [8]. A meta-analysis conducted by Racine and colleagues [9] on 29 studies stated that 1 in 4 youth globally is experiencing clinically elevated depression symptoms, while 1 in 5 youth is experiencing clinically elevated anxiety symptoms. In addition, the study revealed that youth mental health difficulties during the COVID-19 pandemic have likely doubled compared to pre-pandemic estimates.

Several studies reported reduced well-being, life satisfaction, and increased rates of internalizing and externalizing problems in children and adolescents during the first wave [10,11,12,13]. Among a large sample of children and adolescents aged 1–19 years old, from 2.2% to 9.9% of them reported emotional and behavioral problems above the clinical cutoff, and between 15.3% and 43.0% stated an increase in these problems during the pandemic [10].

Similar results arose from recent Italian studies, showing the high prevalence of emotional problems [14,15,16,17,18], as well as the disruption of daily routines and sleep–wake schedules [16,17,19] during the national quarantine in children and adolescents.

In a recent review [20] confirming the high prevalence of psychological distress in children and adolescents, the authors highlighted some sociodemographic risk factors that seem to exacerbate the negative effects of the COVID-19 pandemic on mental health in youth. For instance, a few studies underlined age-related differences in several emotional and behavioral problems. Older children and adolescents reported more depressive, anxious [13,21,22,23,24] and stress symptoms [25] than younger ones. Furthermore, in an Italian study [26] aimed at assessing the perceived risk related to COVID-19 and the psychological experience of quarantine in a large sample of adolescents aged between 13 and 20 years, regression analysis showed that older and females adolescents reported more negative feelings than younger and males ones. Contrasting results emerged from a recent cross-sectional study [10] comparing the effects of the COVID-19 pandemic on mental health in three age groups: 1–6 years, 7–10 years, and 11–19 years. The authors found age-related differences in the type and frequency of problems reported. While preschool children (1–6 years) had the largest increase in oppositional behaviors, adolescents reported the largest increase in emotional problems. Also, compared to preschool and school-aged children, adolescents experienced a significantly larger decrease in emotional and behavioral problems during the pandemic [10]. Accordingly, in two studies it emerged that, although COVID-19 affected adolescents’ emotions and lifestyles, they showed a good ability to manage the situation [27,28].

Besides age, gender is another factor frequently explored in the analysis of the general mental health state during the pandemic in youth. Several studies reported gender differences in psychological distress during COVID-19, with youth females exhibiting more affective symptoms [18,21,22,23,25,29], higher decrease in life satisfaction [12,18] and less self-confidence [26] than males. Instead, no gender differences in psychological distress were found in children aged between 7 and 11 [24] and between 6 and 14 years old [17].

During subsequent waves of the pandemic, while the majority of work activities have been totally restored, school life has struggled to get back to normal in Italy, alternating between continuous openings, closures, and shifting from face-to-face to distance learning. This was especially true for high school students, who have been forced to continue with distance teaching for 50%, or even in full in the so-called “red zones”. Since school has a protective role for physical and mental health in youth [30] and adolescence is a vulnerable life period due to developmental tasks teenagers have to face [31], it is conceivable that high school students might have paid the highest price of the pandemic.

The current study aims at assessing this hypothesis by administering an online survey to a large sample of students aged between 8 and 19 years old, living in the South of Italy. Specifically, we explored students’ perceived psychological distress, using an ad hoc online questionnaire administered during the partial lockdown of the third wave in Italy, with a focus on age and gender differences. Consistent with previous research, we expect that older and females students would report higher psychological distress compared to younger and males students and that psychological distress is associated with age and the type of education provided (face-to-face or distance learning).

## 2. Materials and Methods

### 2.1. Participants and Procedure

We analyzed data collected by the Order of Psychologists of 3 regions of the South of Italy: Campania, Sicilia and Calabria. The Order of Psychologists is a public institution that has the function of supervising the activities of psychologists from all regions of Italy. These functions are carried out at both national and regional levels. Here, the Order of Psychologists of Campania initially drew up the study protocol and extended it to the regional boards of psychologists in Sicily and Calabria regions. The institutions involved then made contact with primary and secondary schools of each respective region, through advertisements on research-related websites and social media groups, in order to disseminate the study protocol to students and their parents. The target population was students aged between 8 and 19 years who were living in the South of Italy. The sample was a convenience sample. That is, the schools interested in participating in the study arranged a meeting with the parents of their students, during which the objectives of the study, information on the processing of personal data, and on the administration procedure were shared. Parents who gave their consent to participate in the study received the link to access the survey and extended it to their sons.

The anonymous online survey was administered from March to July 2021 through the SurveyMonkey’s platform.

A total of 1064 participants (males = 368) completed the survey. Participants were all volunteers, and they were asked to respond to a set of general sociodemographic questions as well as to ad hoc questions about psychological well-being.

The survey took about 3 min to be completed.

Participants were deidentified and data were protected by an unauthorized access according to SurveyMonkey’s privacy policy.

All subjects gave their informed consent for inclusion before they participated in the study. The study was conducted in accordance with the Declaration of Helsinki, and the protocol was approved by the Ethics Committee of Neapolisanit s.r.l. (Project identification code: NEA_RS_04112020_ETIC).

Additional information about the survey is reported in Appendix A, according to the Checklist for Reporting Results of Internet E-Surveys (CHERRIES; [32]).

### 2.2. Survey Structure

The first author (Dr. Angelo Rega) created the survey, together with the team of psychologists of the research and intervention group in the school psychology of the Order of Psychologists of Campania. The survey consists of three sections: an informed consent statement, sociodemographic questions about age, gender, and education, and the assessment of the psychological distress. The latter was made through 21 questions about negative thoughts and emotions, anxiety, depression, tiredness, stress, sleep disorders and dysfunctional coping strategies (e.g., avoidance of specific situations or social isolation).

Students were asked to answer each question by rating how often each symptom was experienced in the last months on a 3-point scale from “never” to “often/frequently”. Answers to each question were collapsed to obtain a total score representing the psychological distress, with higher scores indicating higher levels of psychological distress.

### 2.3. Data Analysis

The first set of descriptive analyses was carried out on the demographic characteristics of our sample. Then, we tested the internal consistency of the questionnaire by Cronbach’s alpha. The structure of the scale was evaluated by carrying out a principal component analysis (PCA) with VARIMAX rotation. The eigenvalue >1 criterion was applied to identify the number of independent components, since its interpretation is less subjective and arbitrary as compared to the scree plot criterion. The internal consistency and the PCA were carried out including the entire set of items (N = 21).

Due to the non-normal distribution of variables (Shapiro–Wilk test: W = 0.885, *p* < 0.001), non-parametric statistics were chosen for data analysis.

Differences among elementary, middle, and high school students (hence “type of school”) in psychological distress were assessed through the Kruskal–Wallis H test. The Mann–Whitney U test was used to evaluate gender differences in psychological distress in the whole sample and then separately for each type of school.

Furthermore, we computed a new nominal variable, namely the type of education provided (face-to-face or distance learning). This variable was codified consistently with the national measures planned for school settings to contain the spread of the contagion (see https://temi.camera.it/leg18/temi/le-misure-adottate-a-seguito-dell-emergenza-coronavirus-covid-19-per-il-mondo-dell-istruzione-scuola-istruzione-e-formazione-professionale-universit-istituzioni-afam.html, accessed on 20 December 2021).

Finally, we sought to explore the extent to which the type of education intervened between the independent variable “Age” and the dependent variable “Psychological distress”. A mediation analysis has been carried out. Following MacKinnon et al. ([33]; see also [34]), we performed two regression analyses: (1) the first analysis with “Age” as independent and “Psychological distress” as a dependent variable; (2) the second with “Age” and “Psychological distress” as independent variables and “Type of education” as the dependent variable. To test the consistency of mediation analysis, we carried out a further regression analysis in order to verify whether the coefficient of “Age” on “Psychological distress” got smaller when the Type of education was included.

All data were analyzed using the Statistical Package for Social Sciences (SPSS) version 27 (IBM, Armonk, NY, USA).

## 3. Results

### 3.1. Demographics

All participants completed the survey; however, five participants were excluded from the analysis due to missing data, resulting in a final sample of 1059 students. Demographic information of the sample is shown in Table 1. The majority of respondents were females (65.3%), aged between 16 and 19 years (41.9%), and attended high schools (56.6%).

### 3.2. Internal Consistency and Principal Component Analysis

The internal consistency analysis reported a Cronbach’s alpha of 0.97, showing an excellent reliability of the survey items (see [35]). The PCA revealed one eigenvalue exceeding 1, accounting for 62.85% of variance. In other words, the analysis showed a one-factor model, which provided the best fit. PCA’s results confirmed our choice to collapse the individuals’ responses in order to compute one total score reflecting psychological distress levels.

### 3.3. Type of School Differences in Psychological Distress

Significant differences in psychological distress total score emerged for the three school levels (elementary: median = 62, middle: median = 70, high: median = 127; Kruskal–Wallis H = 457.07, *p* < 0.001). As shown in Table 2, all post hoc comparisons revealed significant differences between all student groups (*p* < 0.001), reflecting a linear increase in total score in older students relative to younger ones (elementary < middle, *p* < 0.001; elementary < high, *p* < 0.001, middle < high, *p* < 0.001). These comparisons are shown in Figure 1.

### 3.4. Gender Differences in Psychological Distress

The analysis carried out on the entire sample revealed a main effect of gender in terms of psychological distress (males: median= 84.5, females: median= 111; Mann–Whitney U = 155,482.00, *p* < 0.001). These results were confirmed only for high school students when the same analysis was carried out for each type of school separately (males: median = 111, females: median = 135; Mann–Whitney U = 49,818.50, *p* < 0.001). No within-group differences were found between males and females in elementary (males: median = 62, females: median = 62; Mann–Whitney U = 7877.50, *p* = 0.46) and middle students (median = 67.5, females: median = 71; Mann–Whitney U = 5845.50, *p* = 0.15).

Figure 2 displays psychological distress total scores in males and females for the three school levels.

### 3.5. Mediatory Effect of Type of Education on the Relation between Age and Psychological Distress

We designed a mediation model to test the mediator effect of type of education (face-to-face or distance learning) on the link between the age of our participants and the psychological distress. Consistently with MacKinnon et al. [33] and Baron and Kenny [34], we performed three analyses: (1) a regression analysis with “Age” as independent and “Psychological distress” as the dependent variable; (2) two regression analyses with “Age” and “Psychological distress” as independent variables and “Type of education” as dependent variable (see Figure 3 for a graphic representation of the results). The first regression reported a causal relationship between the age of participants and the psychological distress (β = 0.68; *p* < 0.001), with older students reporting higher psychological distress scores. The second regression analysis showed, as expected, that the age of participants was related to the type of education provided: older students were more likely to be involved in distance learning (β = 0.92; *p* < 0.001). Additionally, results of the third regression reported a significant relation between the type of education and psychological distress (β = 0.65; *p* < 0.001): distance learning was associated with higher psychological distress. All regression analysis results are summarized in Table 3.

Finally, a regression model was designed to test whether the coefficient of “Age” on “Psychological distress” got smaller when “Type of education” was included. The results showed a smaller coefficient of the independent variable Age, when the type of education was included in the model (respectively β = 0.68, β = 0.48). Both *p* values were <0.001.

Bound together, these results suggested that the type of education mediates the link between age of students and psychological distress.

## 4. Discussion

The COVID-19 pandemic represents a traumatic and stressful event that the whole world has been fighting for almost two years. Italy was the first European country facing the pandemic through a range of strict measures of containment (i.e., total lockdown), which became more eased during subsequent waves (i.e., partial lockdown). Although useful to prevent the spread of the infection, these disease control strategies had a negative impact on mental health in the general Italian population [5,6,36,37]. As for children and adolescents, while there are data on the psychological effects of total lockdown, findings collected during subsequent waves of the pandemic are very scarce. To our knowledge, this is the first study exploring perceived psychological distress during the partial lockdown in a large sample of Italian students, with a focus on possible age and gender differences in emotional and behavioral responses.

As expected, we found a linear increase in psychological distress in older students relative to younger ones. In particular, psychological distress was higher in middle school students compared to elementary ones, and in high school students in comparison with the other two groups. These results are in line with other studies showing that older adolescents experienced significantly larger increases in emotional and behavioral problems [13,22,23,24,26]. Adolescence is a vulnerable life period due to the developmental tasks teenagers have to face, including the establishment of a sense of mastery, identity, and intimacy as well as pubertal and hormonal changes [31]. Previous research has also shown that the prevalence of internalizing disorders increases from early to mid-late adolescence [38,39]. Therefore, the higher psychological distress we found in older students may be in part due to developmental factors. This hypothesis is supported by our mediation analysis showing a causal relationship between students’ age and their psychological distress.

Also, it is plausible that the COVID-19 pandemic may have added further stress to their already vulnerable status, by reducing social contacts, peer support, and school connectedness, which represent protective factors for adolescents’ mental health [40,41]. In contrast, some authors claimed that most adolescents have proven to be resilient during the first wave of the pandemic, showing abilities to rebuild their habits and social networks [27,28], with a potentially positive impact on their self-efficacy and mental health [10]. Although the emergency experienced in the first lockdown may have fostered the development of resilience resources, mental concerns may surface later in development [17,42], possibly triggered by subsequent waves of the pandemic, when the uncertain situation became chronic and unstable, at least for older adolescents. Indeed, the rapid disruption of their daily routines, due to frequent school openings and closures and the continuous shift from face-to-face to distance learning, may have made it challenging for adolescents to adapt to the situation. In fact, although distance learning has proven to be useful during the lockdown, face-to-face learning is fundamental in promoting dialogue, involvement, and human contact [43]. The hypothesis of a protective role of attending school in person is in line with our mediation analysis, suggesting that online learning may have added further burden to the increased psychological distress already explained by developmental factors.

We found significant gender differences in psychological distress in the total sample, which accounted for higher scores in females than males in high school students. This finding is in line with several studies that consistently pointed out that being female represents a risk factor for mental health during the lockdown [12,18,21,22,23,25,26,29]. This may be in part explained by the fact that girls are generally more prone to internalizing problems than boys (for reviews see [44,45]) and that sex differences in emotional and stress reactions increased from childhood to adolescence [46,47,48]. In addition, it has been shown that girls rely more on social support networks as a coping strategy in challenging situations [49]. Therefore, transitions to distance learning and reduced school and social connectedness, along with biological factors, can both explain our finding of robust gender differences in high school girls but not in younger ones. This explanation is consistent with previous studies showing no gender differences in children and younger adolescents during the lockdown [17,24].

Several limitations suggest caution in interpreting our results. First, given the cross-sectional nature of the study, we did not collect baseline records, so we cannot ascertain whether distress actually changed compared to the first wave or even to the pre-pandemic period. Longitudinal studies are needed to evaluate psychological distress fluctuations among the pandemic waves. Moreover, our survey was built ad hoc, it does not allow us to define cutoff points, useful to understand if the participants’ scores are placed above or below a predetermined risk threshold. Future studies should assess this gap by comparing students’ levels of psychological distress with normative data to track down those at risk and in need of specific psychological support. Another caveat is related to the lack of specific information about students’ perceptions of the COVID-19 emergency, as well as the consequent changes in their daily lives due to the lockdown, which imposes caution in interpreting our mediation analysis results. In fact, the lack of this information prevents us from determining the impact of lockdown on psychological distress in our sample. Nevertheless, the emerged strong relationship between psychological distress and distance learning suggests rather an important role of face-to-face schooling that should be taken into account when choosing Coronavirus prevention strategies.

Finally, we suggest being cautious in extending the results from our sample to the general population. The distribution of the survey through research-related websites and social media groups is usually related to self-selection bias. In our study, the online survey did not allow us to recruit all the Southern Italian students. Only students from Campania, Calabria and Sicilia completed the survey, while data from the other two Regions (Puglia and Basilicata) were not attained. With this type of sampling, the generalizability of results is limited to populations that share similar characteristics with the sample. In other words, our sample might be not representative of the whole population of Southern Italian students, but to those who can be reached by an online survey and living in the three regions mentioned above.

Notwithstanding its limitations, this study has several strengths. First, as already mentioned, this is the first study addressing psychological distress in Southern Italian children and adolescents during the third wave of the pandemic. Another important strength concerned the very large sample size, allowing us to capture information about more than one thousand students and directly from them.

## 5. Conclusions

The present study explored gender and age differences in psychological distress in a large sample of Italian Southern students during the third wave of the pandemic. Consistent with previous studies, we found that being female and older adolescent represent risk factors for students’ mental well-being. In addition, we highlight the importance of face-to-face schooling as a protective factor in youth’s psychological well-being. Future longitudinal studies are needed to better understand the mental health trajectory during the pandemic and to develop clinical practices and public health strategies targeting vulnerable populations. Indeed, in a period of continuous school openings and closures, it is therefore necessary to take this aspect into account, choose the most appropriate prevention measures to tackle the COVID-19 infection spread and safeguard people’s psychological well-being.

## Figures and Tables

**Figure 1 ijerph-19-05532-f001:**
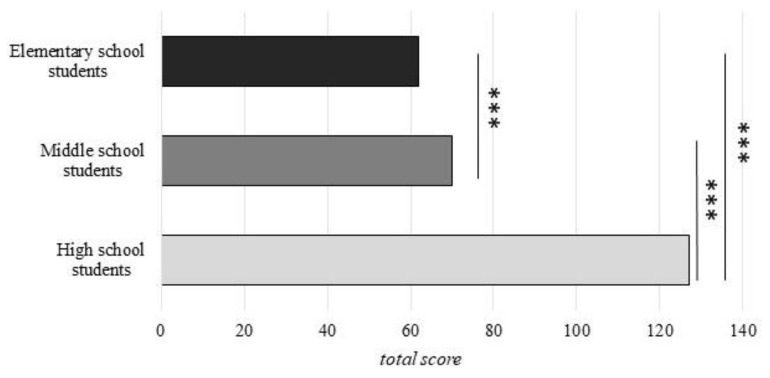
Psychological distress total score among elementary, middle and high school students. Significant between-group comparisons are reported. Data are presented as median. *** *p* < 0.001.

**Figure 2 ijerph-19-05532-f002:**
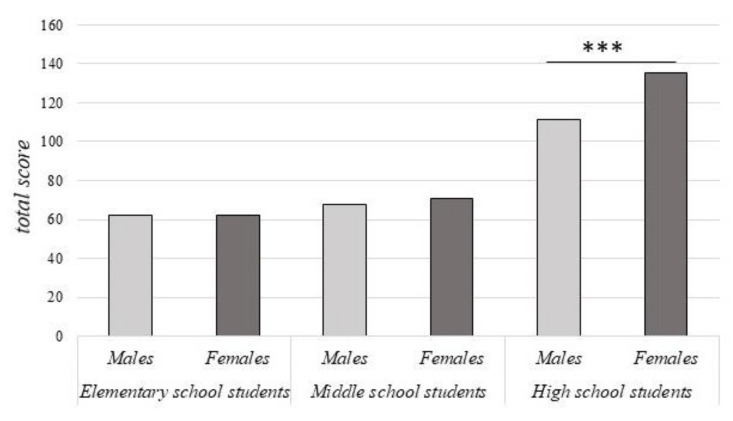
Gender differences in psychological distress total score among elementary, middle and high students. Significant within-group comparisons are reported. Data are presented as median. *** *p* < 0.001.

**Figure 3 ijerph-19-05532-f003:**
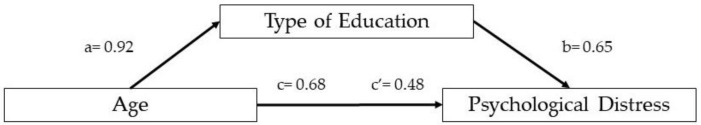
The mediation effects of type of education on the psychological distress. In the arrows are reported the β values about the effect of the age on the type of education (a), the effect of type of education on psychological distress (b), the total effect of the age on the psychological distress (c), and the direct effect of the age on the psychological distress controlling for the mediator. All *p* values were <0.001.

**Table 1 ijerph-19-05532-t001:** Demographic characteristics of the sample.

Age Groups	N	%
8–10 years	232	21.89
11–12 years	171	16.13
13–15 years	212	20.00
16–19 years	444	41.89
School levels	N	%
Elementary	249	23.49
Middle	210	19.81
High	600	56.60
Gender	N	%
Males	368	34.72
Females	391	65.19

**Table 2 ijerph-19-05532-t002:** Pairwise comparisons among elementary, middle and high school students.

Pairwise Comparisons	Mann–Whitney U	*p*-Value
Elementary-Middle	−123.96	*p* < 0.001
Elementary-High	−450.02	*p* < 0.001
Middle-High	−326.06	*p* < 0.001

Notes: All *p* values were adjusted using Bonferroni correction.

**Table 3 ijerph-19-05532-t003:** Results of the three regression analysis: (a) with “Psychological distress” as dependent variable; (b) with “Age” and “Psychological distress” as independent variables and “Type of education” as the dependent variable.

**(a)**	Beta	t	*p*-Value
Age	0.65	25.74	<0.001
**(b)**	Beta	t	*p*-Value
Age	0.92	74.39	<0.001
Psychological distress	0.65	24.31	<0.001

## Data Availability

The data presented in this study are available on request from the corresponding author. The data are not publicly available due to privacy and ethical reasons.

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
