# Peer review of "Age-Related Differences in Psychological Distress during the COVID-19 Pandemic"

_ijerph, 2022, doi:10.3390/ijerph19095532_

Round 1

Reviewer 1 Report

Thank you for sending this paper through for review, it was interesting and timely and the sample size was very good. The paper looks at psychological distress likely related to COVID and whether age and gender of children is related to levels of distress.

Major comments:

  • You mention throughout that your findings indicate that lockdowns are likely responsible for the distress kids are feeling, however, your data does not reflect this. From what I can tell the data indicates that distance education is related to distress scores - you can report this but I don't think your data allows you to link lockdowns to your results unless you asked specifically about lockdowns in your questionnaire
  • There are a few grammatical errors throughout, the paper would need a proof read by a native English speaker
  • More detail is needed in the aims of the study - perhaps outlining that the questionnaire is to do with psychological distress and that you aim to determine if psychological distress is lower than norms? 

Minor Comments:

  • Use of subjective language like 'huge' and 'twist our life' should be omitted from the paper
  • Check citation requirements for page 2 line 55 - most referencing systems use et al. or colleagues - not their coworkers
  • You will need to explain what the order of psychologists is, people from outside of Italy do not know this
  • Line 115 - is 'M' male?
  • Was the psychological distress questionnaire created by the authors?
  • Figure one should say elementary school and high school students

Author Response

Dear Reviewer,

Kind regards,

Mariangela Cerasuolo

Reviewer 2 Report

The present article is relevant and interesting and provides new evidence on age and gender variables associated with psychological well-being related to the COVID 19 pandemic of students of different age groups in Southern Italy.

Reviewing the paper I find some points that should be addressed by the authors

  1. Abstract. Revise the wording in relation to the description of the age of the participants since they refer to M=368 and it is not congruent with the data presented.

Material and Method: 

Participants, it does not refer who signed the informed consent since being minors, parents or guardians should have given informed consent.

Measuring instrument: when designing an instrument, it is important to mention that the psychometric properties of the instrument were evaluated and   Cronbach´s alpha and exploratory factor analysis are not enough; the external validity analysis would be missing. Kindly refer to the checklist for Reporting Results of Internet E-Surveys (CHERRIES) and include it in your method section.

  1. Results:

Revise in the wording and graphics to unify the number of decimals. 

Item 3.2: It is suggested to argue why the external validity of an instrument designed by the authors to evaluate psychological well-being was not performed.

3.3. It is suggested to add the description of the cut-off points to be able to identify the level of psychological distress of the participants and to present a table that presents the comparison by school grade. Figure 1 is insufficient to describe the results.

3.5 The authors refer to review graph 45 but it does not exist.

Figure 3 presents a mediation analysis (type of education) between the independent variable (age) and the dependent variable (psychological distress). The authors refer that 2 regressions were performed and three are described.   It is suggested to elaborate a table to visualize the B indices and significance in relation to the variables analyzed in the regressions. (age, gender, type of education).

Author Response

(The authors gave the same response as above.)
